# Emergent Communication in a Multi-Modal, Multi-Step Referential Game

**Katrina Evtimova**[1], **Andrew Drozdov**[2], **Douwe Kiela**[3], and **Kyunghyun Cho**[1,2,3,4]

[1]Center for Data Science. New York University
[2]Department of Computer Science. New York University
[3]Facebook AI Research
[4]CIFAR Azrieli Global Scholar

## Abstract

Inspired by previous work on emergent communication in referential games, we propose a novel multi-modal, multi-step referential game, where the sender and receiver have access to distinct modalities of an object, and their information exchange is bidirectional and of arbitrary duration. The multi-modal multi-step setting allows agents to develop an internal communication significantly closer to natural language, in that they share a single set of messages, and that the length of the conversation may vary according to the difficulty of the task. We examine these properties empirically using a dataset consisting of images and textual descriptions of mammals, where the agents are tasked with identifying the correct object. Our experiments indicate that a robust and efficient communication protocol emerges, where gradual information exchange informs better predictions and higher communication bandwidth improves generalization.

## 1 Introduction

Recently, there has been a surge of work on neural network-based multi-agent systems that are capable of communicating with each other in order to solve a problem. Two distinct lines of research can be discerned. In the first one, communication is used as an essential tool for sharing information among multiple active agents in a reinforcement learning scenario (Sukhbaatar et al., 2016; Foerster et al., 2016; Mordatch & Abbeel, 2017; Andreas et al., 2017). Each of the active agents is, in addition to its traditional capability of interacting with the environment, able to communicate with other agents. A population of such agents is subsequently jointly tuned to reach a common goal. The main goal of this line of work is to use communication (which may be continuous) as a means to enhance learning in a difficult, sparse-reward environment. The communication may also mimic human conversation, e.g., in settings where agents engage in natural language dialogue based on a shared visual modality (Das et al., 2017; Strub et al., 2017).

In contrast, the goal of our work is to *learn* the communication protocol, and aligns more closely with another line of research, which focuses on investigating and analyzing the emergence of communication in (cooperative) multi-agent referential games (Lewis, 2008; Skyrms, 2010; Steels & Loetzsch, 2012), where one agent (the sender) must communicate what it sees using some discrete emergent communication protocol, while the other agent (the receiver) is tasked with figuring out what the first agent saw. These lines of work are partially motivated by the idea that artificial communication (and other manifestations of machine intelligence) can emerge through interacting with the world and/or other agents, which could then converge towards human language (Gauthier & Mordatch, 2016; Mikolov et al., 2015; Lake et al., 2016; Kiela et al., 2016). (Lazaridou et al., 2016) have recently proposed a basic version of this game, where there is only a single transmission of a message from the sender to the receiver, as a test bed for both inducing and analyzing a communication protocol between two neural network-based agents. A related approach to using a referential game with two agents is proposed by (Andreas & Klein, 2016). (Jorge et al., 2016) have more recently introduced a game similar to the setting above, but with multiple transmissions of messages between

the two agents. The sender is, however, strictly limited to sending single bit (yes/no) messages, and the number of exchanges is kept fixed.

These earlier works lack two fundamental aspects of human communication in solving cooperative games. First, human information exchange is bidirectional with symmetric communication abilities, and spans exchanges of arbitrary length. In other words, linguistic interaction is not one-way, and can take as long or as short as it needs. Second, the information exchange emerges as a result of a disparity in knowledge or access to information, with the capability of bridging different modalities. For example, a human who has never seen a tiger but knows that it is a "big cat with stripes" would be able to identify one in a picture without effort. That is, humans can identify a previously unseen object from a textual description alone, while agents in previous interaction games have access to the same modality (a picture) and their shared communication protocol.

Based on these considerations, we extend the basic referential game used in (Lazaridou et al., 2016; Andreas & Klein, 2016; Jorge et al., 2016) and (Havrylov & Titov, 2017) into a *multi-modal, multi-step referential game*. Firstly, our two agents, the sender and receiver, are grounded in different modalities: one has access only to the visual modality, while the other has access only to textual information (**multi-modal**). The sender sees an image and communicates it to the receiver whose job is to determine which object the sender refers to, while only having access to a set of textual descriptions. Secondly, communication is bidirectional and symmetrical, in that both the sender and receiver may send an arbitrary binary vector to each other. Furthermore, we allow the receiver to autonomously decide when to terminate a conversation, which leads to an adaptive-length conversation (**multi-step**). The multi-modal nature of our proposal enforces symmetric, high-bandwidth communication, as it is not enough for the agents to simply exchange the carbon copies of their modalities (e.g. communicating the value of an arbitrary pixel in an image) in order to solve the problem. The multi-step nature of our work allows us to train the agents to develop an efficient strategy of communication, implicitly encouraging a shorter conversation for simpler objects and a longer conversation for more complex objects.

We evaluate and analyze the proposed multi-modal, multi-step referential game by creating a new dataset consisting of images of mammals and their textual descriptions. The task is somewhat related to recently proposed multi-modal dialogue games, such as that of (de Vries et al., 2016), but then played by agents using their own emergent communication. We build neural network-based sender and receiver, implementing techniques such as visual attention (Xu et al., 2015) and textual attention (Bahdanau et al., 2014). Each agent generates a multi-dimensional binary message at each time step, and the receiver decides whether to terminate the conversation. We train both agents jointly using policy gradient (Williams, 1992).

## 2 Multi-Modal, Multi-Step Referential Game

**Game** The proposed multi-modal, multi-step referential **game** is characterized by a tuple

$$G = \langle S, O, O_S, O_R, s^* \rangle.$$

$S$ is a set of all possible messages used for communication by both the sender and receiver. An analogy of $S$ in natural languages would be a set of all possible sentences. Unlike (Jorge et al., 2016), we let $S$ be shared between the two agents, which makes the proposed game a more realistic proxy to natural language conversations where two parties share a single vocabulary. In this paper, we define the set of symbols to be a set of $d$-dimensional binary vectors, reminiscent of the widely-used bag-of-words representation of a natural language sentence. That is, $S = \{0, 1\}^d$.

$O$ is a set of objects. $O_S$ and $O_R$ are the sets of two separate views, or modes, of the objects in $O$, exposed to the sender and receiver, respectively. Due to the variability introduced by the choice of mode, the cardinalities of the latter two sets may differ, i.e., $|O_S| \neq |O_R|$, and it is usual for the cardinalities of both $O_S$ and $O_R$ to be greater than or equal to that of $O$, i.e., $|O_S| \geq |O|$ and $|O_R| \geq |O|$. In this paper, for instance, $O$ is a set of selected mammals, and $O_S$ and $O_R$ are, respectively, images and textual descriptions of those mammals: $|O_S| \gg |O_R| = |O|$.

The ground-truth map between $O_S$ and $O_R$ is given as

$$s^* : O_S \times O_R \to \{0, 1\}.$$

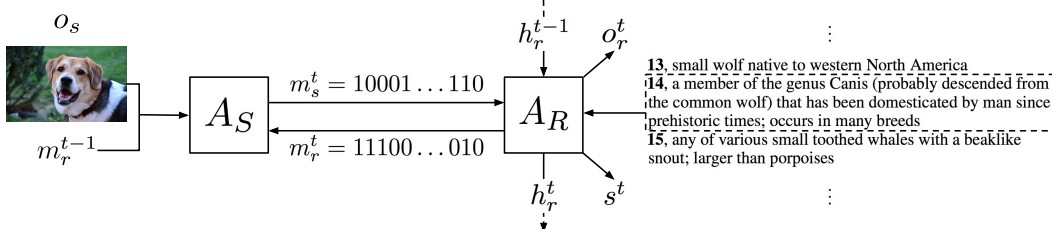

Figure 1: Visualizing a sender-receiver exchange at time step $t$. See Sec. 2 and 3 for more details.

This function $s^*$ is used to determine whether elements $o_s \in O_S$ and $o_r \in O_R$ belong to the same object in $O$. It returns $1$ when they do, and $0$ otherwise. At the end of a conversation, the receiver selects an element from $O_R$ as an answer, and $s^*$ is used as a scorer of this particular conversation based on the sender's object $o_s$ and the receiver's prediction $\hat{o}_r$.

**Agents**    The proposed game is played between two agents, sender $A_S$ and receiver $A_R$. A **sender** is a stochastic function that takes as input the sender's view of an object $o_s \in O_S$ and the message $m_r \in S$ received from the receiver and outputs a binary message $m_s \in S$. That is,

$$A_S : O_S \times S \to S.$$

We constrain the sender to be memory-less in order to ensure any message created by the sender is a response to an immediate message sent by the receiver.

Unlike the sender, it is necessary for the receiver to possess a memory in order to reason through a series of message exchanges with the sender and make a final prediction. The receiver also has an option to determine whether to terminate the on-going conversation. We thus define the **receiver** as:

$$A_R : S \times \mathbb{R}^q \to \Xi \times O_R \times S \times \mathbb{R}^q,$$

where $\Xi = \{0, 1\}$ indicates whether to terminate the conversation. It receives the sender's message $m_s \in S$ and its memory $h \in \mathbb{R}^q$ from the previous step, and stochastically outputs: (1) whether to terminate the conversation $s \in \{0, 1\}$, (2) its prediction $\hat{o}_r \in O_R$ (if decided to terminate) and (3) a message $m_r \in S$ back to the sender (if decided not to terminate).

**Play**    Given $G$, one game instance is initiated by uniformly selecting an object $o$ from the object set $O$. A corresponding view $o_s \in O_S$ is sampled and given to the sender $A_S$. The whole set $O_R$ is provided to the receiver $A_R$. The receiver's memory and initial message are learned as separate parameters.

## 3    AGENTS

At each time step $t \in \{1, \ldots, T_{\max}\}$, the sender computes its message $m_s^t = A_S(o_s, m_r^{t-1})$. This message is then transmitted to the receiver. The receiver updates its memory $h_r^t$, decides whether to terminate the conversation $s^t$, makes its prediction $o_r^t$, and creates a response: $(s^t, o_r^t, m_r^t, h_r^t) = A_R(m_s^t, h_r^{t-1})$. If $s^t = 1$, the conversation terminates, and the receiver's prediction $o_r^t$ is used to score this game instance, i.e., $s^*(o_s, o_r^t)$. Otherwise, this process repeats in the next time step: $t \leftarrow t + 1$. Fig. 1 depicts a single sender-receiver exchange at time step $t$.

**Feedforward Sender**    Let $o_s \in O_S$ be a real-valued vector, and $m_r \in S$ be a $d$-dimensional binary message. We build a sender $A_S$ as a feedforward neural network that outputs a $d$-dimensional factorized Bernoulli distribution. It first computes the hidden state $h_s$ by

$$h_s = f_s(o_s, m_r), \tag{1}$$

and computes $p(m_{s,j} = 1)$ for all $j = 1, \ldots, d$ as

$$p(m_{s,j} = 1) = \sigma(w_{s,j}^\top h_s + b_{s,j}),$$

where $\sigma$ is a sigmoid function, and $w_{s,j} \in \mathbb{R}^{\dim(h_s)}$ and $b_{s,j} \in \mathbb{R}$ are the weight vector and bias, respectively. During training, we sample a sender's message from this distribution, while during test time we take the most likely message, i.e., $m_{s,j} = \arg\max_{b \in \{0,1\}} p(m_{s,j} = b)$.

**Attention-based Sender** When the view $o_s$ of an object is given as a set of vectors $\{o_{s_1}, \ldots, o_{s_n}\}$ rather than a single vector, we implement and test an attention mechanism from (Bahdanau et al., 2014; Xu et al., 2015). For each vector in the set, we first compute the attention weight against the received message $m_r$ as $\alpha_j = \frac{\exp(f_{s,\text{att}}(o_{s_j}, m_r))}{\sum_{j'=1}^{n} \exp(f_{s,\text{att}}(o_{s_{j'}}, m_r))}$, and take the weighted-sum of the input vectors: $\tilde{o}_s = \sum_{j=1}^{n} \alpha_j o_{s_j}$. This weighted sum is used instead of $o_s$ as an input to $f_s$ in Eq. (1). Intuitively, this process of attention corresponds to selecting a subset of the sender's view of an object according to a receiver's query.

**Recurrent Receiver** Let $o_r \in O_R$ be a real-valued vector, and $m_s \in S$ be a $d$-dimensional binary message received from the sender. A receiver $A_R$ is a recurrent neural network that first updates its memory by $h_r^t = f_r(m_s^t, h_r^{t-1}) \in \mathbb{R}^q$, where $f_r$ is a recurrent activation function. We use a gated recurrent unit (GRU, Cho et al., 2014). The initial message from the receiver to the sender, $m_r^0$, is learned as a separate parameter.

Given the updated memory vector $h_r^t$, the receiver first computes whether to terminate the conversation. This is done by outputting a stop probability, as in

$$p(s^t = 1) = \sigma(w_{r,s}^\top h_r^t + b_{r,s}),$$

where $w_{r,s} \in \mathbb{R}^q$ and $b_{r,s} \in \mathbb{R}$ are the weight vector and bias, respectively. The receiver terminates the conversation ($s^t = 1$) by either sampling from (during training) or taking the most likely value (during test time) of this distribution. If $s^t = 0$, the receiver computes the message distribution similarly to the sender as a $d$-dimensional factorized Bernoulli distribution:

$$p(m_{r,j}^t = 1) = \sigma(w_{r,j}^\top \tanh\left(W_r^\top h_r^t + U_r^\top \Big(\sum_{o_r \in O_R} p(o_r = 1) g_r(o_r)\Big) + c_r\right) + b_{r,j}),$$

where $g_r : \mathbb{R}^{\dim(o_r)} \to \mathbb{R}^q$ is a trainable function that embeds $o_r$ into a $q$-dimensional real-valued vector space. The second term inside the $\tanh$ function ensures that the message generated by the receiver takes into consideration the receiver's current belief $p(o_r = 1)$ (see Eq. (2)) on which object the sender is viewing.

If $s^t = 1$ (terminate), the receiver instead produces its prediction by computing the distribution over all the elements in $O_R$:

$$p(o_r = 1) = \frac{\exp(g_r(o_r)^\top h_r^t)}{\sum_{o_r' \in O_R} \exp(g_r(o_r')^\top h_r^t)}. \tag{2}$$

Again, $g_r(o_r)$ is the embedding of an object $o$ based on the receiver's view $o_r$, similarly to what was proposed by (Larochelle et al., 2008). The receiver's prediction is given by $\hat{o}_r = \arg\max_{o_r \in O_R} p(o_r = 1)$, and the entire prediction distribution is used to compute the cross-entropy loss.

**Attention-based Receiver** Similarly to the sender, we can incorporate the attention mechanism in the receiver. This is done at the level of the embedding function $g_r$ by modifying it to take as input both the set of vectors $o_r = \{o_{r,1}, \ldots, o_{r,n}\}$ and the current memory vector $h_r^t$. Attention weights over the view vectors are computed against the memory vector, and their weighted sum $\tilde{o}_r$, or its affine transformation to $\mathbb{R}^q$, is returned.

## 4 TRAINING

Both the sender and receiver are jointly trained in order to maximize the score $s^*(o_s, \hat{o}_r)$. Our per-instance loss function $L^i$ is the sum of the classification loss $L_c^i$ and the reinforcement learning loss $L_r^i$. The classification loss is a usual cross-entropy loss defined as

$$L_c^i = \log p(o_r^* = 1),$$

where $o_r^* \in O_R$ is the view of the correct object. The reinforcement learning loss is defined as

$$L_r^i = \underbrace{\sum_{t=1}^{T} (R - B_s(o_s, m_r^{t-1})) \sum_{j=1}^{d} \log p(m_{s,j}^t)}_{\text{sender}} + \underbrace{(R - B_r(m_r^t, h_r^{t-1}))(\log p(s^t) + \sum_{j=1}^{d} \log p(m_{r,j}^t))}_{\text{receiver}},$$

where $R$ is a reward given by the ground-truth mapping $s^*$. This reinforcement learning loss corresponds to REINFORCE (Williams, 1992). $B_s$ and $B_r$ are baseline estimators for the sender and receiver, respectively, and both of them are trained to predict the final reward $R$, as suggested by (Mnih & Gregor, 2014):

$$L_B^i = \sum_{t=1}^{T} (R - B_s(o_s, m_r^{t-1}))^2 + (R - B_r(m_s^t, h_r^{t-1}))^2.$$

In order to facilitate the exploration by the sender and receiver during training, we regularize the negative entropies of the sender's and receiver's message distributions. We also minimize the negative entropy of the receiver's termination distribution to encourage the conversation to be of length $1 - (\frac{1}{2})^{T_{\max}}$ on average.

The final per-instance loss can then be written as

$$L^i = L_c^i + L_r^i - \sum_{t=1}^{T} \left( \lambda_s H(s^t) + \lambda_m \sum_{j=1}^{d} (H(m_{s,j}^t) + H(m_{r,j}^t)) \right),$$

where $H$ is the entropy, and $\lambda_s \geq 0$ and $\lambda_m \geq 0$ are regularization coefficients. We minimize this loss by computing its gradient with respect to the parameters of both the sender and receiver and taking a step toward the opposite direction.

We list all the mathematical symbols used in the description of the game in Appendix A.

## 5 EXPERIMENTAL SETTINGS

### 5.1 DATA COLLECTION AND PREPROCESSING

We collect a new dataset consisting of images and textual descriptions of mammals. We crawl the nodes in the subtree of the "mammal" synset in WordNet (Miller, 1995). For each node, we collect the word $o$ and the corresponding textual description $o_r$ in order to construct the object set $O$ and the receiver's view set $O_R$. For each word $o$, we query Flickr to retrieve as many as 650 images [1]. These images form the sender's view set $O_S$.

We sample 70 mammals from the subtree and build three sets from the collected data. First, we keep a subset of sixty mammals for training (550 images per mammal) and set aside data for validation (50 images per mammal) and test (20 images per mammal). This constitutes the **in-domain test**, that measures how well the model does on mammals that it is familiar with. We use the remaining ten mammals to build an **out-of-domain test** set (100 images per mammal), which allows us to test the generalization ability of the sender and receiver to unseen objects, and thereby to determine whether the receiver indeed relies on the availability of a different mode from the sender.

In addition to the mammals, we build a third test set consisting of 10 different types of insects, rather than mammals. To construct this **transfer test**, we uniformly select 100 images per insect at random from the ImageNet dataset (Deng et al., 2009), while the descriptions are collected from WordNet, similarly to the mammals. The test is meant to measure an extreme case of zero-shot generalization, to an entirely different category of objects (i.e., insects rather than mammals, and images from ImageNet rather than from Flickr).

**Image Processing** Instead of a raw image, we use features extracted by ResNet-34 (He et al., 2016). With the attention-based sender, we use 64 ($8 \times 8$) 512-dimensional feature vectors from the final convolutional layer. Otherwise, we use the 512-dimensional feature vector after average pooling those 64 vectors. We do not fine-tune the network.

---

[1] We query Flickr, obtaining more than 650 images per word, then we remove duplicates and use a heuristic to discard undesirables images. Duplicates are detected using dHash (Tantos, 2017). As a heuristic, we take an image classifier that was trained on ImageNet (Krizhevsky et al., 2012), classify each candidate image, and discard an image if its most likely class is not an animal. We randomly select from the remaining images to acquire the desired amount.

**Text Processing**    Each description is lowercased. Stopwords are filtered using the Stopwords Corpus included in NLTK (Bird et al., 2009). We treat each description as a bag of unique words by removing any duplicates. The average description length is 9.1 words with a standard deviation of 3.16. Because our dataset is relatively small, especially in the textual mode, we use pretrained 100-dimensional GloVe word embeddings (Pennington et al., 2014). With the attention-based receiver, we consider a set of such GloVe vectors as $o_r$, and otherwise, the average of those vectors is used as the representation of a description.

## 5.2    Models and Training

**Feedforward Sender**    When attention is not used, the sender is configured to have a single hidden layer with 256 tanh units. The input $o_s$ is constructed by concatenating the image vector, the receiver's message vector, their point-wise difference and point-wise product, after embedding the image and message vectors into the same space by a linear transformation. The attention-based sender uses a single-layer feedforward network with 256 tanh units to compute the attention weights.

**Recurrent Receiver**    The receiver is a single hidden-layer recurrent neural network with 64 gated recurrent units. When the receiver is configured to use attention over the words in each description, we use a feedforward network with a single hidden layer of 64 rectified linear units.

**Baseline Networks**    The baseline networks $B_s$ and $B_r$ are both feedforward networks with a single hidden layer of 500 rectified linear units each. The receiver's baseline network takes as input the recurrent hidden state $h_r^{t-1}$ but does not backpropagate the error gradient through the receiver.

**Training and Evaluation**    We train both the sender and receiver as well as associated baseline networks using RMSProp (Tieleman & Hinton, 2012) with learning rate set to $10^{-4}$ and minibatches of size 64 each. The coefficients for the entropy regularization, $\lambda_s$ and $\lambda_m$, are set to $0.08$ and $0.01$ respectively, based on the development set performance from the preliminary experiments. Each training run is early-stopped based on the development set accuracy for a maximum of 500 epochs. We evaluate each model on a test set by computing the accuracy@$K$, where K is set to be 10% of the number of categories in each of the three test sets (K is either 6 or 7, since we always include the classes from training). We use this metric to enable comparison between the different test sets and to avoid overpenalizing predicting similar classes, e.g. kangaroo and wallaby. We set the maximum length of a conversation to be 10, i.e., $T_{\max} = 10$. We train on a single GPU (Nvidia Titan X Pascal), and a single experiment takes roughly 8 hours for 500 epochs.

**Code**    We used PyTorch [http://pytorch.org]. Our implementation of the agents and instructions on how to build the dataset are available on Github [https://github.com/nyu-dl/MultimodalGame].

## 6    Results and Analysis

The model and approach in this paper are differentiated from previous work mainly by: 1) the variable conversation length, 2) the multi-modal nature of the game and 3) the particular nature of the communication protocol, i.e., the messages. In this section, we experimentally examine our setup and specifically test the following hypotheses:

- The more difficult or complex the referential game, the more dialogue turns would be needed if humans were to play it. Similarly, we expect the receiver to need more information, and ask more questions, if the problem is more difficult. Hence, we examine **the relationship between conversation length and accuracy/difficulty**.

- As the agents take turns in a continuing conversation, more information becomes available, which implies that the receiver should become more sure about its prediction, even if the problem is difficult to begin with. Thus, we separately examine **the confidence of predictions as the conversation progresses**.

- The agents play very different roles in the game. On the one hand, we would hypothesize the receiver's messages to become more and more specific. For example, if the receiver has already established that the picture is of a feline, it does not make sense to ask, e.g., whether

the animal has tusks or fins. This implies that the entropy of its messages should decrease. On the other hand, as questions become more specific, they are also likely to become more difficult for the sender to answer with high confidence. Answering that something is an aquatic mammal is easier than describing, e.g., the particular shape of a fin. Consequently, the entropy of the sender's messages is likely to increase as it grows less confident in its answers. To examine this, we analyze **the information theoretic content of the messages** sent by both agents.

In what follows, we discuss experiments along the lines of these hypotheses. In addition, we analyze the impact of changing the message dimensionality, and the effect of applying visual and linguistic attention mechanisms.

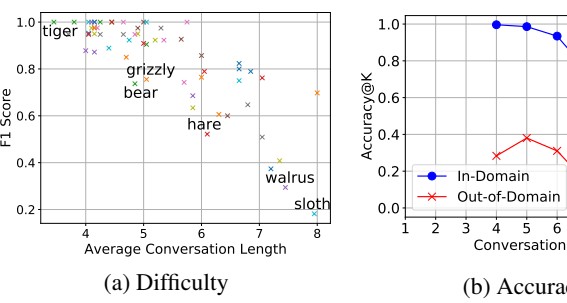

(a) Difficulty    (b) Accuracy

Figure 2: (a) Difficulty (measured by F1) versus conversation length across classes. A negative correlation is observed, implying that difficult classes require more turns. (b) Accuracy@$K$ versus conversation length for the in-domain (blue) and out-of-domain (red) test sets.

**Conversation length and accuracy/difficulty** We train a pair of agents with an adaptive conversation length in which the receiver may terminate the conversation early based on the stop probability. Once training is done, we inspect the relationship between average conversation length and difficulty across classes, as well as the accuracy per the conversation length by partitioning the test examples into length-based bins.

We expect that more difficult classes require a higher average length of exchange. To test this hypothesis, we use the accuracy of a *separate* classifier as a proxy for the difficulty of a sample. Specifically, we train a classifier based on a pre-trained ResNet-50, in which we freeze all but the last layer, and obtain the F1 score per class evaluated on the in-domain test set. The Pearson correlation between the F1 score and average conversation length across classes is $-0.81$ with a $p$-value of $4 \times 10^{-15}$ implying a statistically significant negative relationship, as displayed in Fig. 2 (a).

In addition, we present the accuracies against the conversation lengths (as automatically determined by the receiver) in Fig. 2 (b). We notice a clear trend with the in-domain test set: examples for which the conversations are shorter are better classified, which might indicate that they are easier. It is important to remember that the receiver's stop probability is not artificially tied to the performance nor confidence of the receiver's prediction, but is simply *learned* by playing the proposed game. A similar trend can be observed with the out-of-domain test set, however, to a lesser degree. A similar trend of having longer conversation for more difficult objects is also found with humans in the game of 20 questions (Cohen & Lake, 2016).[2]

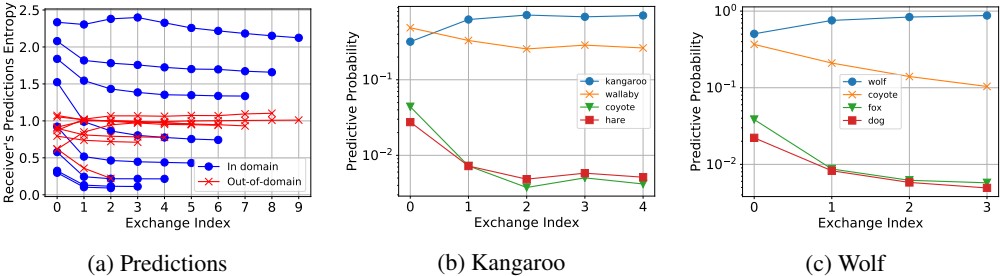

(a) Predictions      (b) Kangaroo      (c) Wolf

Figure 3: (a) Prediction entropy over the conversation using the in-domain (blue) and out-of-domain (red) test sets. (b, c) Prediction certainty over time in example conversations about Kangaroo and Wolf, respectively.

---

[2] Accuracy scores in relation to the number of questions were obtained via personal communication.

**Conversation length and confidence**    With the agents trained with an adaptive conversation length, we can investigate how the prediction uncertainty of the receiver evolves over time. We plot the evolution of the entropy of the prediction distribution in Fig. 3 (a) averaged per conversation length bucket. We first notice that the conversation length, determined by the receiver on its own, correlates well with the prediction confidence (measured as negative entropy) of the receiver. Also, it is clear on the in-domain test set that the entropy almost monotonically decreases over the conversation, and the receiver terminates the conversation when the predictive entropy converges. This trend is however not apparent with the out-of-domain test set, which we attribute to the difficulty of zero-shot generalization.

The goal of the conversation, i.e., the series of message exchanges, is to distinguish among many different objects. The initial message from the sender could for example give a rough idea of the high-level category that an object belongs to, after which the goal becomes to distinguish different objects within that high-level category. In other words, objects in a single such cluster, which are visually similar due to the sender's access to the visual mode of an object, are predicted at different time steps in the conversation.

We qualitatively examine this hypothesis by visualizing how the predictive probabilities of the receiver evolve over a conversation. In Fig. 3 (b,c), we show two example categories – kangaroo and wolf. As the conversation progress and more information is gathered for the receiver, similar but incorrect categories receive smaller probabilities than the correct one. We notice a similar trend with all other categories.

**Information theoretic message content**    In the previous section, we examined how prediction certainty evolved over time. We can do the same with the messages sent by the respective agents. In Fig. 4, we plot the entropies of the message distributions by the sender and receiver. We notice that, as the conversation progresses, the entropy decreases for the receiver, while it increases for the sender. This observation can be explained by the following conjecture. As the receiver accumulates information transmitted by the sender, the set of possible queries to send back to the sender shrinks, and consequently the entropy decreases. It could be said that the questions become more specific as more information becomes available to the receiver as it *zones in* on the correct answer. On the other hand, as the receiver's message becomes more specific and difficult to answer, the certainty of the sender in providing the correct answer decreases, thereby increasing the entropy of the sender's message distribution. We notice a similar trend on the out-of-domain test set as well.

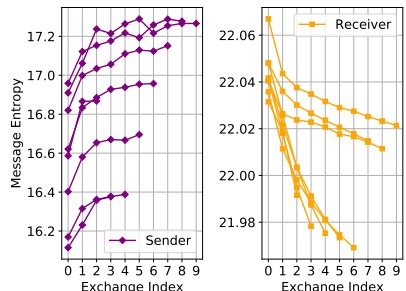

Figure 4: Message entropy over the conversation on the in-domain test set of the sender (left) and receiver (right).

**Effect of the message dimensionality**    Next, we vary the dimensionality $d$ of each message to investigate the impact of the constraint on the communication channel, while keeping the conversation length adaptive. We generally expect a better accuracy with a higher bandwidth. More specifically, we expect the generalization to unseen categories (out-of-domain test) would improve as the information bandwidth of the communication channel increases. When the bandwidth is limited, the agents will be forced to create a communication protocol highly specialized for categories seen during training. On the other hand, the agents will learn to decompose structures underlying visual and textual modes of an object into more generalizable descriptions with a higher bandwidth channel.

The accuracies reported in Fig. 5 agree well with this hypothesis. On the in-domain test set, we do not see significant improvement nor degradation as the message

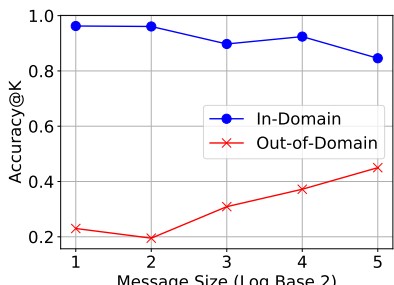

Figure 5:    Accuracy@$K$ on the In-Domain ($K = 6$) and Out-of-Domain ($K = 7$) test sets for the Adaptive models of varying message size. We notice the increasing accuracy on the out-of-domain test set as the bandwidth of the channel increases.

dimensionality changes. We observe, however, a strong correlation between the message dimensionality and the accuracy on the out-of-domain test set. With 32-dimensional messages, the agents were able to achieve up to 45% accuracy@7 on the out-of-domain test set which consists of 10 mammals not seen during training. The effect of modifying the message dimension was less clear when measured against the transfer set.

**Effect of Attention Mechanism**    All the experiments so far have been run without attention mechanism. We train additional three pairs of agents with 32-dimensional message vectors; (1) attention-based sender, (2) attention-based receiver, and (3) attention-based sender and attention-based receiver. On the in-domain test set, we are not able to observe any improvement from the attention mechanism on either of the agents. We did however notice that the attention mechanism (attention-based receiver) significantly improves the accuracy on the transfer test set from 16.9% up to 27.4%. We conjecture that this is due to the fact that attention allows the agents to focus on the aspects of the objects (e.g. certain words in descriptions; or regions in images) that they are familiar with, which means that they are less susceptible to the noise introduced from being exposed to an entirely new category. We leave further analysis of the effect of the attention mechanism for future work.

**Is communication necessary?**    One important consideration is whether the trained agents utilize the adaptability of the communication protocol. It is indeed possible that the sender does not learn to shape communication and simply relies on the random communication protocol decided by the random initialization of its parameters. In this case, the receiver will need to recover information from the sender sent via this random communication channel.

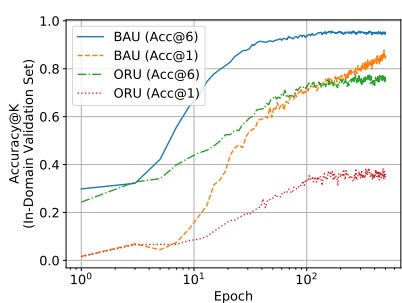

In order to verify this is not the case, we train a pair of agents without updating the parameters of the sender. As the receiver is still updated, and the sender's information still flows toward the receiver, learning happens. We, however, observe that the overall performance significantly lags behind the case when agents are trained together, as shown in Fig. 6. This suggests that the agents must learn a

Figure 6: Learning curves when both agents are updated (BAU), and only the receiver is updated (ORU).

new, task-specific communication protocol, which emerges in order to solve the problem successfully.[3]

## 7    CONCLUSION

In this paper, we have proposed a novel, multi-modal, multi-step referential game for building and analyzing communication-based neural agents. The design of the game enables more human-like communication between two agents, by allowing a variable-length conversation with a symmetric communication. The conducted experiments and analyses reveal three interesting properties of the communication protocol, or artificial language, that emerges from learning to play the proposed game.

First, the sender and receiver are able to adjust the length of the conversation based on the difficulty of predicting the correct object. The length of the conversation is found to (negatively) correlate with the confidence of the receiver in making predictions. Second, the receiver gradually asks more specific questions as the conversation progresses. This results in an increase of entropy in the sender's message distribution, as there are more ways to answer those highly specific questions. We further observe that increasing the bandwidth of communication, measured in terms of the message dimensionality, allows for improved *zero-shot* generalization. Most importantly, we present a suite of hypotheses and associated experiments for investigating an emergent communication protocol, which we believe will be useful for the future research on emergent communication.

**Future Direction**    Despite the significant extension we have made to the basic referential game, the proposed multi-modal, multi-step game also exhibits a number of limitations. First, an emergent

---

[3]There are additional statistics about the stability of training in Appendix B.

communication from this game is not entirely symmetric as there is no constraint that prevents the two agents from partitioning the message space. This could be addressed by having more than two agents interacting with each other while exchanging their roles, which we leave as future work. Second, the message set $S$ consists of fixed-dimensional binary vectors. This choice effectively prevents other linguistic structures, such as syntax. Third, the proposed game, as well as any existing referential game, does not require any action, other than speaking. This is in contrast to the first line of research discussed earlier in Sec. 1, where communication happens among active agents. We anticipate a future research direction in which both of these approaches are combined.

### ACKNOWLEDGMENTS

We thank Brenden Lake and Alex Cohen for valuable discussion. We also thank Maximilian Nickel, Y-Lan Boureau, Jason Weston, Dhruv Batra, and Devi Parikh for helpful suggestions. KC thanks for support by AdeptMind, Tencent, eBay, NVIDIA, and CIFAR. AD thanks the NVIDIA Corporation for their donation of a Titan X Pascal. This work is done by KE as a part of the course DS-GA 1010-001 Independent Study in Data Science at the Center for Data Science, New York University. A part of Fig. 1 is licensed from EmmyMik/CC BY 2.0/https://www.flickr.com/photos/emmymik/8206632393/.

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

## A  TABLE OF NOTATIONS

Table 1: Table of Notations

| Symbol | DESCRIPTION |
|---|---|
| $A_S$ | sender agent |
| $A_R$ | receiver agent |
| $S$ | set of all possible messages used for communication by both agents |
| $O$ | set of mammal classes |
| $O_S$ | set of mammal images available to the sender |
| $O_R$ | set of mammal descriptions available to the receiver |
| $s^*$ | ground-truth map between $O_S$ and $O_R$, namely $s^* \colon O_S \times O_R \to \{0,1\}$ |
| $o_s$ | element of $O_S$ |
| $o_r$ | element of $O_R$ |
| $o_r^*$ | element of $O_R$ corresponding to the correct object in a sender-receiver exchange |
| $o_r^t$ | the receiver's predicted distribution over objects in $O_R$ at timestep $t$ |
| $\hat{o}_r$ | the receiver's prediction |
| $m_s$ | binary message sent by the sender |
| $m_r$ | binary message sent by the receiver |
| $\Xi$ | set of binary indicators for terminating a conversation $\{0,1\}$ |
| $s$ | value of indicator for terminating conversation yielded by the receiver |
| $s^t$ | value of indicator for terminating conversation yielded by the receiver at time step $t$ |
| $T_{max}$ | maximal value for number of time steps in a conversation |
| $t$ | time step in conversation between sender and receiver |
| $m_s^t$ | binary message generated by sender at time step $t$ |
| $m_r^t$ | binary message generated by receiver at time step $t$ |
| $h_s$ | hidden state vector of the sender |
| $h_r$ | hidden state vector of the receiver |
| $h_r^t$ | hidden state of receiver at time step $t$ |
| $f_s(o_s, m_r)$ | function computing hidden state $h_s$ of sender |
| $f_{s,att}(o_s, m_r)$ | function computing hidden state $h_s$ of attention-based sender |
| $f_r(m_s, h_r^{t-1})$ | the receiver's recurrent activation function computing $h_r^t$ |
| $B_s$ | baseline feedforward network of the sender |
| $B_s$ | baseline feedforward network of the receiver |
| $m_{s,j}$ | the $j$-th coordinate of the sender's message |
| $w_{s,j}$ | the $j$-th column of the sender's weight matrix |
| $b_{s,j}$ | the $j$-th coordinate of the sender's bias vector |
| $g_r(o_r)$ | embedding of an object $o$ by the receiver's view $o_r$ |
| $m_{r,j}^t$ | the $j$-th coordinate of the receiver's message |
| $W_r$ | the receiver's weight matrix for its hidden space |
| $U_r$ | the receiver's weight matrix for embeddings of $o_r \in O_R$ |
| $c_r$ | the receiver's bias vector for embeddings of $o_r \in O_R$ |
| $w_{r,j}$ | the $j$-th column of the receiver's weight matrix $W_r$ |
| $b_{r,j}$ | the $j$-th coordinate of the receiver's bias vector for hidden state |
| $v^\top$ | the transpose of vector $v$ |
| $L^i$ | per-instance loss |
| $L_R^i$ | per-instance reinforcement learning loss |
| $L_B^i$ | per-instance baseline loss |
| $R$ | reward from ground-truth mapping $s^*$ |
| $H$ | entropy |
| $\lambda_m$ | entropy regularization coefficient for the binary messages distributions of both agents |
| $\lambda_s$ | entropy regularization coefficient for the receiver's termination distribution |

## B  STABILITY OF TRAINING

We ran our standard setup[4] six times using different random seeds. For each experiment, we trained the model until convergence using early stopping against the validation data, then measured the loss and accuracy on the in-domain test set. The accuracy@6 had mean of $96.6\%$ with variance of $1.98e{-}1$, the accuracy@1 had mean of $86.0\%$ with variance $7.59e{-}1$, and the loss had mean of $0.611$ with variance $2.72e{-}3$. These results suggest that the model is not only effective at classifying images, but also robust to random restart.

---

[4]The standard setup uses adaptive conversation lengths with a maximum length of 10 and message dimension of 32. The values of other hyperparameters are described in Section 5.2.

