# OpenReview forum: "Emergent Communication in a Multi-Modal, Multi-Step Referential Game"
_ICLR.cc/2018/Conference — Accept (Poster)_

### Official Review · AnonReviewer2 · 2017-11-27

**Rating:** 7
**Confidence:** 4

**Review:**

--------------
Summary and Evaluation:
--------------

The paper presents a nice set of experiments on language emergence in a mutli-modal, multi-step setting. The multi-modal reference game provides an interesting setting for communication, with agents learning to map descriptions to images. The receiving agent's direct control over dialog length is also novel and allows for the interesting analysis presented in later sections.

Overall I think this is an interesting and well-designed work; however, some details are missing that I think would make for a stronger submission (see weaknesses).


--------------
Strengths:
--------------
- Generally well-written with the Results and Analysis section appearing especially thought-out and nicely presented.

- The proposed reference game provides a number of novel contributions -- giving the agents control over dialog length, providing both agents with the same vocabulary without constraints on how each uses it (implicit through pretraining or explicit in the structure/loss), and introducing an asymmetric multi-modal context for the dialog.

- The analysis is extensive and well-grounded in the three key hypothesis presented at the beginning of Section 6.

--------------
Weaknesses:
--------------

- There is room to improve the clarity of Sections 3 and 4 and I encourage the authors to revisit these sections. Some specific suggestions that might help:
		- numbering all display style equations
		- when describing the recurrent receiver, explain the case where it terminates (s^t=1) first such that P(o_r=1) is defined prior to being used in the message generation equation.

- I did not see an argument in support of the accuracy@K metric. Why is putting the ground truth in the top 10% the appropriate metric in this setting? Is it to enable comparison between the in-domain, out-domain, and transfer settings?

- Unless I missed something, the transfer test set results only comes up once in the context of attention methods and are not mentioned elsewhere. Why is this? It seems appropriate to include in Figure 5 if no where else in the analysis.

- Do the authors have a sense for how sensitive these results are to different runs of the training process?

- I did not understand this line from Section 5.1: "and discarding any image with a category beyond the 398-th most frequent one, as classified by a pretrained ImageNet classifier'"

- It is not specified (or I missed it) whether the F1 scores from the separate classifier are from training or test set evaluations.

- I would have liked to see analysis on the training process such as a plot of reward (or baseline adjusted reward) over training iterations.

- I encourage authors to see the EMNLP 2017 paper "Natural Language Does Not Emerge ‘Naturally’ in Multi-Agent Dialog" which also perform multi-round dialogs between two agents. Like this work, the authors also proposed removing memory from one of the agents as a means to avoid learning degenerate 'non-dialog' protocols.

- Very minor point: the use of fixed-length, non-sequence style utterances is somewhat disappointing given the other steps made in the paper to make the reference game more 'human like' such as early termination, shared vocabularies, and unconstrained utterance types. I understand however that this is left as future work.


--------------
Curiosities:
--------------
- I think the analysis is Figure 3 b,c is interesting and wonder if something similar can be computed over all examples. One option would be to plot accuracy@k for different utterance indexes -- essentially forcing the model to make a prediction after each round of dialog (or simply repeating its prediction if the model has chosen to stop).

---

> ### Author Response · Authors · 2018-01-03
> **Response to Reviewer**
>
> We would like to thank the reviewer for their thoughtful compliments and criticism. In particular, the detailed list of areas for improvement have lead us to run additional experiments and make edits in the text that we believe have strengthened our work.
>
> Let us address your concerns and questions below.
>
> > analysis on the training process
>
> We’ve updated Figure 6 in the paper to display Accuracy@1 in addition to Accuracy@6. We hope this metric plotted over each epoch gives a useful overview of the training process and some insight in how the model’s performance changes over time.
>
> > Do the authors have a sense for how sensitive these results are to different runs of the training process?
>
> We ran six experiments with different random seeds and reported the mean and variance on their loss and accuracy in Appendix B, but would be open to include these values in the main text if this seems useful.
>
> > the transfer test set
>
> There was not much to be gleaned from the transfer set besides the effect of the attention mechanism. We’re more explicit about saying so in Section 6.
>
> > the accuracy@K metric
>
> We use this metric since many mammal classes are quite similar to each other, and we don't want to overpenalize predicting similar classes such as kangaroo and wallaby. As suggested by the reviewer, this metric also enables comparison between the in-domain, out-domain, and transfer test sets.
>
> > the F1 scores from the separate classifier are from training or test set evaluations
>
> The plot in Figure 2a and its associated F1 scores are derived from the in-domain test set.
>
> > discarding any image with a category beyond the 398-th most frequent one
>
> When we build our dataset, we discard images that are not likely to be an animal, as determined by a pre-trained classifier.
>
> > numbering all display style equations
>
> We appreciate the reviewer’s suggestion to add equation numbers, but believe that since we have so many equations, it is alright to only number the equations that we reference explicitly in the text.
>
> > when describing the recurrent receiver, explain the case where it terminates (s^t=1) first such that P(o_r=1) is defined prior to being used in the message generation equation
>
> The first message of the receiver is learned as a separate parameter in all cases and we’ve mentioned this in the “Recurrent Receiver” portion of Section 3.
>
> > the analysis is Figure 3 b,c
>
> For Figure 3b and 3c, we show only the top-4 predicted classes because the probabilities given to the other classes are negligible in comparison. The observation that we made regarding this figure (that as the conversation progresses, similar but incorrect categories receive smaller probabilities than the correct one) held for all other categories, but we limited to these two classes as we felt this sufficiently conveyed the idea.
>
> > I encourage authors to see the EMNLP 2017 paper "Natural Language Does Not Emerge ‘Naturally’ in Multi-Agent Dialog" which also perform multi-round dialogs between two agents. Like this work, the authors also proposed removing memory from one of the agents as a means to avoid learning degenerate 'non-dialog' protocols.
>
> And
>
> > Very minor point: the use of fixed-length, non-sequence style utterances is somewhat disappointing given the other steps made in the paper to make the reference game more 'human like' such as early termination, shared vocabularies, and unconstrained utterance types. I understand however that this is left as future work.
>
> There are some matters that we will leave for future work. Kottur et al. explain how limiting memory can force consistency over different steps in a dialog. This can be a useful property, but our work was primarily concerned with the distribution over messages and the model’s prediction confidence. It’s a natural progression to investigate the meaning of these messages as a follow-up work, and to attempt models that encode meaning not only in individual words, but also the latent structure in sequences of words.

---

### Official Review · AnonReviewer3 · 2017-11-27
**Interesting paper that extends basic referential game**

**Rating:** 7
**Confidence:** 4

**Review:**

The paper proposes a new multi-modal, multi-step reference game, where the sender has access to visual data and the receiver has access to textual messages, and also the conversation can be terminated by the receiver when proper.

Later, the paper describes their idea and extension in details and reports comprehensive experiment results of a number of hypotheses. The research questions seems straightforward, but it is good to see those experiments review some interesting points.  One thing I am bit concerned is that the results are based on a single dataset. Do we have other datasets that can be used?

The authors also lay out further several research directions. Overall, I think this paper is easy to read and good.

---

> ### Author Response · Authors · 2018-01-03
> **Response to Reviewer**
>
> We’d like to thank the reviewer for their thoughtful feedback. In response to the following comment:
>
> > One thing I am bit concerned is that the results are based on a single dataset.
>
> A distinguishing property of our dataset is that, in addition to images, each class has an informative textual description, and there is a natural hierarchy of properties shared between classes. As there wasn’t a similar dataset already available, we had to collect the data ourselves. In section 5.2 of the de-anonymized version of the paper, we’ll include a link to our codebase which contains instructions to build such a dataset.

---

### Official Review · AnonReviewer1 · 2017-11-28
**Interesting take on representation learning**

**Rating:** 7
**Confidence:** 3

**Review:**

The setup in the paper for learning representations is different to many other approaches in the area, using to agents that communicate over descriptions of objects using different modalities. The experimental setup is interesting in that it allows comparing approaches in learning an effective representation. The paper does mention the agents will be available, but leaves open wether the dataset will be also available. For reproducibility and comparisons, this availability would be essential.

I like that the paper gives a bit of context, but presentation of results could be clearer, and I am missing some more explicit information on training and results (eg how long / how many training examples, how many testing, classification rates, etc).
The paper says is the training procedure is described in Appendix A, but as far as I see that contains the table of notations.

---

> ### Author Response · Authors · 2018-01-03
> **Response to Reviewer**
>
> Thank you for your thoughtful comments.
>
> > The paper does mention the agents will be available, but leaves open whether the dataset will be also available.
>
> You bring up a great point that in order to reproduce our results it would be necessary to have access to a similar dataset. In addition, even with written details of the implementation, it can be difficult to reproduce experiments. For these reasons, we’ve prepared to release the code and instructions on how to build the dataset, and will include a link in the de-anonymized version of the paper. We allude to this in section 5.2 under Code.
>
> > missing some more explicit information on training and results
>
> And
>
> > The paper says the training procedure is described in Appendix A
>
> We also thank the reviewer for pointing out the typo in relation to Appendix A. In terms of training and results, a plot of the classification accuracy by epoch is shown in the updated Figure 6. We added the following details in sections 5.1 and 5.2 that should clear up confusion about the training procedure:
>
> The number of images per class in the out-of-domain test set is 100 images per class (for 10 classes in total).
> We use early stopping with a maximum 500 training epochs.
> We train on a single GPU (Nvidia Titan X Pascal), and a single experiment takes roughly 8 hours for 500 epochs.
>
> Is the addition of these details sufficient?

---

### Decision · Program_Chairs · 2018-01-29
**ICLR 2018 Conference Acceptance Decision**

**Decision:**

Accept (Poster)

**Comment:**

An interesting paper, generally well-written. Though it would be nice to see that the methods and observations generalize to other datasets, it is probably too much to ask as datasets with required properties do not seem to exist.  There is a clear consensus to accept the paper.

+ an interesting extension of previous work on emergent communications (e.g., referential games)
+ well written paper